# Age-Related Sex Differences in Glucose Tolerance by 75 g Oral Glucose Tolerance Test in Japanese

**DOI:** 10.3390/nu14224868

**Published:** 2022-11-17

**Authors:** Akihiro Yoshida, Takao Kimura, Katsuhiko Tsunekawa, Yoshifumi Shoho, Yoshimaro Yanagawa, Osamu Araki, Tomoyuki Aoki, Takayuki Ogiwara, Masami Murakami

**Affiliations:** 1Department of Clinical Laboratory Medicine, Gunma University Graduate School of Medicine, Showa-machi 3-39-22, Meabshi 371-8511, Gunma, Japan; 2Clinical Laboratory Center, Gunma University Hospital, Showa-machi 3-39-22, Meabshi 371-8511, Gunma, Japan; 3Center for Food Science and Wellness, Gunma University, Aramaki-machi 4-2, Meabshi 371-8510, Gunma, Japan; 4Department of Sports Education, Faculty of Education, Ikuei University, Kyome-machi 1656-1, Takasaki 370-0011, Gunma, Japan; 5Internal Medicine, Division of Endocrinology and Diabetes, Saiseikai Maebashi Hospital, Kamishindenmachi, Maebashi 371-0821, Gunma, Japan

**Keywords:** 75 g OGTT, insulin sensitivity, insulin resistance, age, young, middle-aged, sex difference, lipid, HOMA-IR, HOMA-β

## Abstract

To elucidate the age-related sex difference in glucose tolerance, we conducted 75 g oral glucose tolerance tests in 1156 participants. Participants were divided into four groups, namely, young (22–29) males, young females, middle-aged (>50) males, and middle-aged females. According to the Japanese Clinical Practice Guideline for Diabetes 2019, the prevalence of normal glucose tolerance (NGT) was significantly lower in middle-aged than in young participants. The prevalence of high-normal fasting plasma glucose (FPG) was higher, and NGT was lower in young males (high-normal FPG 15.2%, NGT 82.0%) than young females (high-FPG 3.9%, NGT 94.3%). Combined glucose intolerance (CGI) was higher and NGT was lower in middle-aged males (CGI 10.2%, NGT 25.2%) than in middle-aged females (CGI 3.3%, NGT 39.8%). FPG and body mass index (BMI) were the lowest and Homeostatic model assessment beta cell function (HOMA-β) was the highest in young females, followed by young males, middle-aged females, and middle-aged males. Multiple linear regression analysis revealed that BMI weakly correlated with HOMA-β and Matsuda index in all subjects except young females. The superior glucose tolerance in females was apparent in young, but attenuated in middle-aged females. The differences are due to the higher insulin secretion potential and lower BMI in young females.

## 1. Introduction

The increasing incidence of diabetes is a serious issue worldwide. According to the International Diabetes Federation Diabetes Atlas, in 2021, there were 537 million people with diabetes worldwide and this is expected to increase to 643 million by 2030 [1]. Abnormalities in glucose metabolism including impaired glucose tolerance (IGT) and type 2 diabetes mellitus (T2DM) can exist with fasting plasma glucose (FPG) of <6.1 mmol/L [2]. Among Japanese, the decrease in insulin sensitivity and secretion begins in patients with impaired fasting glucose (IFG) 5.6–6.1 mmol/L [3]. Moreover, Japanese individuals with FPG of >5.6 mmol/L have an increased 5-year incidence rate of T2DM [4].

Aging is one of the major risk factors for T2DM. Middle-aged individuals show higher insulin secretion than younger ones [5], whereas homeostatic model assessment beta cell function (HOMA-β), a simple index of insulin secretion, declines with age [6]. In Japanese males, age-related FPG elevation is primarily caused by decreased β-cell function rather than increased insulin resistance [7]. A slight impairment in insulin secretion initially starts in individuals with normal glucose tolerance (NGT) [8]. An 8 year follow-up study of middle-aged Japanese showed that impaired insulin secretion plays a predominant role in the non-obese subtype of non-insulin dependent diabetes mellitus [9].

In addition to IGT due to aging, the sex differences in glucose tolerance have also been pointed out. The National Health and Nutrition Survey in Japan, 2019 by the Japanese Ministry of Health, Labor and Welfare, stated that diabetes is strongly suspected in 19.7% of males and 10.8% of females [10]. The sex difference has been explained by an early-phase insulin release [11] and insulin resistance [12]. Recently, even in young and healthy Japanese individuals within the physiological range of glycemic control, there is a sequential decrease in insulin sensitivity and secretion [13] and cardinal rheological parameters, such as lipid parameters, reflected insulin resistance, and release [14]. Furthermore, we reported a prolonged plasma glucose elevation on 75 g oral glucose tolerance test (75 g OGTT) in young, healthy Japanese individuals [13]. These suggest the possibility that glucose intolerance already begins in healthy youth. Diabetes is more prevalent in males than in females, especially in middle-aged populations [15]. The peak in diabetes prevalence occurs earlier in males than in females and male predominance is specifically observed in middle-aged populations [16]. However, we did not examine sex differences in glucose tolerance of young Japanese. In this study, we conducted 75 g OGTT among young and middle-aged Japanese. We also studied lipid profiles and compared the indices of insulin sensitivity and insulin secretion, since glucose intolerance is associated with dyslipidemia [14,17]. This study was intended to elucidate the sex differences and age-related changes in glucose tolerance through 75 g OGTT.

## 2. Materials and Methods

### 2.1. Participants

As shown in Figure 1A, a total of 1156 individuals participated in this study with written informed consent, of which 785 young Japanese were recruited between May 2010 and November 2018 (505 males and 280 females; aged 22–29 years), and 371 subjects were recruited between March 2007 and November 2011 (127 males and 244 females; aged 50–83 years). No participants were diagnosed with T2DM in the past or received antidiabetic medication. Moreover, 12 (9.4%) of 127 middle-aged males and 48 (19.7%) of 244 older females were taking medications for dyslipidemia.

All the participants signed informed consent forms, and the Gunma University Ethical Review Board for Medical Research Involving Human Subjects (No. 12–41 and 73) approved the study protocol. All ethical and confidentiality considerations were handled in accordance with the Declaration of Helsinki.

### 2.2. Study Design

We performed 75 g OGTT after 12 h fast, measured 0, 30, 60, and 120 min plasma glucose (PG0, PG30, PG60, and PG120, respectively) and insulin levels (IRI0, IRI30, IRI60, and IRI120, respectively), and determined the serum levels of high-density lipoprotein cholesterol (HDL-C), low-density lipoprotein cholesterol (LDL-C), triglyceride (TG), and hemoglobin A1c (HbA1c) at the pre-load time. As shown in Figure 1B and the Appendix A, we classified participants into the diabetes group, combined glucose intolerance (CGI) group IGT group, IFG group, high-normal fasting plasma glucose (high-normal FPG) group, and NGT group according to the Japanese Clinical Practice Guideline for Diabetes 2019 [18].

### 2.3. Laboratory Assays

We measured participants’ heights and weights and calculated body mass index (BMI; weight (kg)/height (m^2^)). Fasting serum HDL-C, LDL-C, and TG concentrations were measured by enzymatic methods with an automatic analyzer (LABOSPECT 008; Hitachi, Tokyo). Serum insulin concentrations were also measured by chemiluminescence immunoassay using an automatic analyzer (AIA-2000 LA; Tosoh, Tokyo). We further measured plasma glucose concentrations by a hexokinase method and HbA1c levels using high-performance liquid chromatography, with automatic analyzers (ADAMS Glucose GA-1170 and ADAMS A1c HA8180, respectively; Arkray, Tokyo, Japan).

### 2.4. Statistical Methods

We calculated areas under the glucose or insulin curves (AUCg and AUCi, respectively) based on the trapezoid rule. We also calculated the homeostasis model assessment of insulin resistance (HOMA-IR), HOMA-β, Matsuda index [19], and insulinogenic index [9] as reported. The insulin secretion/insulin resistance (disposition) index was calculated as the insulinogenic index × Matsuda index [20]. All formulas are listed in the supplement. We calculated an updated HOMA model (HOMA2) [21] using the HOMA2 Calculator provided at https://www.dtu.ox.ac.uk/homacalculator/. HOMA2 cannot be calculated when the insulin concentration is less than 20 pmol/L. HOMA2 in 147 of 1156 participants was not calculable because of the insulin levels < 20 pmol/L. Especially, 89 of the 371 middle-aged participants had insulin levels <20 pmol/L. Therefore, we excluded HOMA2 in this study.

We used the IBM SPSS Statistics version 25 (IBM Corp., Armonk, NY, USA) to perform the statistical analyses. The characteristics of the study participants were expressed as the mean ± standard deviation. Each parameter was compared using Bonferroni’s method, chi-squared test, and one-way analysis of variance followed by Tukey’s post hoc tests. Spearman’s correlation analyses were checked separately for young males, young females, middle-aged males, and middle-aged females. The multiple linear regression model was analyzed to determine the contribution of factors to glycemic parameters separately into young males, young females, middle-aged males, and middle-aged females.

## 3. Results

### 3.1. Comparison of Young Males and Middle-Aged Males

As shown in Table 1, BMI, FPG, AUCg, AUCi, HbA1c, TG, LDL-C, TG/HDL-C ratio, and LDL-C/HDL-C ratio were significantly lower in young males than in middle-aged males. Fasting plasma insulin, HOMA-β, Matsuda index, insulinogenic index, and disposition index were significantly higher in young males than in middle-aged males. Data reflecting these differences is shown in Table 2. The prevalence of NGT was significantly higher in young males than in middle-aged males, whereas the prevalence of high-normal FPG, IFG, IGT, CGI, and diabetes was significantly lower in young males than in middle-aged males. Compared with young males, in middle-aged males, BMI increased and the Matsuda index and HOMA-β decreased, while unexpectedly HOMA-IR decreased. Furthermore, 41 of 505 (8.1%) young males and 32 of 127 (25.2%) middle-aged males had fasting insulin levels <20 pmol/L. Lower fasting insulin levels in middle-aged males resulted in lower HOMA-IR.

### 3.2. Comparison of Young Females and Middle-Aged Females

BMI, FPG, AUCg, HbA1c, TG, LDL-C, TG/HDL-C ratio, and LDL-C/HDL-C ratio were significantly lower in young females than in middle-aged females (Table 1). Fasting plasma insulin, HOMA-β, insulinogenic index, and disposition index were significantly higher in young females than in middle-aged females (Table 1). The prevalence of NGT was significantly higher in young females than in middle-aged females, whereas the prevalence of high-normal FPG, IFG, IGT, CGI, and diabetes was significantly lower in young females than in middle-aged females (Table 2). Compared to young females, in middle-aged females, BMI increased and HOMA-β decreased, while unexpectedly HOMA-IR decreased. 17 of 280 (6.1%) young females and 57 of 244 (23.4%) middle-aged females were fasting insulin levels <20 pmol/L. Lower fasting insulin levels in middle-aged females resulted in lower HOMA-IR.

### 3.3. Comparison of Young Males and Young Females

BMI, FPG, AUCg, HOMA-IR, TG, TG/HDL-C ratio, and LDL-C/HDL-C ratio were significantly lower in young females than in young males. AUCi, HOMA-β, and HDL-C were significantly higher in young females than in young males (Table 1). The prevalence of NGT was significantly higher in young females than in young males, whereas the prevalence of high-normal FPG was significantly lower in young females than in young males (Table 2).

### 3.4. Comparison of Middle-Aged Males and Middle-Aged Females

BMI, FPG, AUCg, TG, TG/HDL-C ratio, and LDL-C/HDL-C ratio were significantly lower in middle-aged females than in middle-aged males (Table 1). The HDL-C level was significantly higher in middle-aged females than in middle-aged males (Table 1). The mean age was slightly higher in middle-aged males than in middle-aged females (Table 1). The prevalence of NGT was significantly higher in middle-aged females than in middle-aged males, whereas the prevalence of CGI was significantly lower in middle-aged females than in middle-aged males (Table 2). As regards the prevalence of diabetes, no significant difference was detected between middle-aged males and females.

### 3.5. Sequential Changes in the Concentrations of Plasma Glucose and Insulin during the 75 gOGTT

PG0, PG30, PG60, PG120, IRI30, IRI 60, and IRI120 were significantly higher in middle-aged males than in young males, whereas IRI0 was significantly lower in middle-aged males than in young males (Figure 2). PG0, PG30, PG60, and PG120 were significantly higher in middle-aged females than in young females, whereas IRI0 and IRI30 were significantly lower in middle-aged females than in young females (Figure 2). PG0, PG30, and PG60 were significantly lower in young females than in young males, whereas IRI30 and IRI120 were significantly higher in young females than in young males (Figure 2). PG0 and PG60 were significantly lower in middle-aged females than in middle-aged males, whereas no significant difference in insulin level was detected during 75 g OGTT between middle-aged females and middle-aged males (Figure 2). Figure 2C shows the difference between the plasma glucose level before and after the glucose load. In young participants, sex differences were observed only at 30 min after glucose load and before load. In middle-aged subjects, sex differences were observed only at 60 min after glucose load and before load. The difference between the plasma glucose level before and after the glucose load was greater in middle-aged subjects than in young subjects. Figure 2D shows the difference between the plasma insulin level before and after the glucose load. In young participants, sex differences were observed only at 30 min after glucose load and before load. In middle-aged subjects, there was no difference between males and females in post- and pre-dose insulin concentrations. The difference between the plasma insulin level 30 min after and before the glucose load was smaller in middle-aged subjects than in young subjects. The difference in insulin concentration between 60 min and preload, and between 120 min and preload, was greater in middle-aged subjects than in young subjects.

Multiple regression analysis is shown in Table 3. The results of Spearman’s correlation analyses are described in the Appendix A. In young males, HOMA-β weakly correlated with BMI; HOMA-IR weakly correlated with BMI, and TG/HDL-C ratio, very weakly correlated with HbA1c and TG; and the Matsuda index very weakly correlated with BMI, TG, and HDL-C. In young females, Insulinogenic index very weakly corelated with HDL-C; HOMA-β very weakly correlated with TG; and Matsuda index very weakly correlated with HbA1c. In middle-aged males, Insulinogenic index moderately correlated with HbA1c and very weakly correlated with TG/HDL-C ratio; HOMA-β weakly correlated with BMI, HbA1c, and TG/HDL-C ratio; and the Matsuda index weakly correlated with BMI and very weakly correlated with HDL-C. In middle-aged females, HOMA-β weakly correlated with BMI, and very weakly correlated with HbA1c, HDL-C, and TG/HDL-C, and the Matsuda index weakly correlated with BMI, and very weakly correlated with HbA1c, and TG/HDL-C.

## 4. Discussion

This study showed the age-related increase in the prevalence of glucose intolerance and Japanese females have better glucose tolerance than Japanese males. In young females, glucose tolerance was clearly better than that of males; however, the difference attenuated in middle-aged individuals. The differences were mainly based on higher insulin secretion potential in females according to HOMA-β and AUCi. PG0, PG60, AUCg, and TG/HDL-C ratio were the lowest in young females, followed by young males, middle-aged females, and middle-aged males. This order appears to be the mirror image of HOMA-β. Among both Japanese middle-aged males and females, HOMA-β is lower and the prevalence of glucose intolerance was higher than those of young males and females. The indices of insulin resistance/secretion weakly correlated with BMI and lipid profiles among individuals, but not among young females. Although lipid markers also reflect glucose tolerance, BMI, fasting insulin, and HOMA-β, which can be calculated from blood glucose levels, are good indicators.

In this study, the higher prevalence of CGI and the lower prevalence of NGT in middle-aged males than middle-aged females; however, middle-aged males and females are equally to have diabetes. This result is in good agreement with previous reports. Diabetes is more prevalent in males than in females, especially in middle-aged populations [15]. The peak in diabetes prevalence occurs earlier in males than in females and male pre-dominance is specifically observed in middle-aged populations [16].

Females had better glucose tolerance than males in both the young and middle-aged participants, whereas the prevalence of glucose intolerance was significantly higher in the middle-aged participants. HOMA-β reflects this fact. HOMA-β declined with age [6,7,22]. Early-phase insulin release deteriorated with age, particularly in females but not insulin sensitivity [11]. This study supports this theory. The difference in insulin concentrations 30 min after and before the glucose load were higher in young females than in males, and were otherwise at similar levels. The difference in plasma glucose levels 30 min after glucose load and before load was lower in young females than in males and was otherwise at the same level. This sex difference disappears in middle-aged males and females. This study supports the theory that HOMA-β in young females was significantly higher than that in young males, whereas no significant difference was observed between middle-aged males and females. Additionally, HOMA-β and insulinogenic indexes were significantly higher in young than in middle-aged participants, whereas AUCi was higher in middle-aged than in young ones. These findings suggest that basal and early postload insulin secretions in middle-aged participants decline due to aging. Middle-aged participants need a longer time for postload insulin secretion to decrease glucose levels than young ones. HOMA-β was significantly higher in young females than in young males, but not in middle-aged males and middle-aged females. These facts suggested the age-related attenuation of sex difference in HOMA-β.

Middle-aged participants had higher fasting insulin levels than younger ones [5]. In contrast, fasting plasma insulin levels among middle-aged participants were significantly lower than those of young participants in this study. Fasting plasma insulin levels were 35 ± 21 pmol/L and the same in 32 middle-aged males classified as NGT and in all middle-aged males. Similarly, fasting plasma insulin levels were 35 ± 23 pmol/L in 97 middle-aged females classified as NGT and 35 ± 21 pmol/L in all middle-aged females. Fasting plasma insulin levels in this study participants were the same as other studies which investigated middle-aged Japanese [3,7]. Therefore, the lower fasting plasma insulin levels in middle-aged participants in this study were characteristic of Japanese or Asian people. These facts could explain the lower fasting plasma insulin levels in middle-aged participants in this study. Many middle-aged Japanese may be at high risk of type 2 diabetes due to a decrease in basal insulin secretion. The lower fasting plasma insulin levels caused lower HOMA-IR in middle-aged participants.

Whole-body insulin sensitivity is higher in females and insulin-stimulated glucose uptake is higher in females skeletal muscle [23]. The postprandial elevation of plasma glucose stimulates insulin secretion and the elevation of plasma insulin stimulates glucose uptake in skeletal muscle leading to the disposal of ingested glucose [24]. Insulin resistance in skeletal muscle begins long before the hyperglycemia becomes evident [24]. In insulin-resistant states, insulin-stimulated glucose uptake in skeletal muscle was markedly impaired [24]. The lower the skeletal muscle mass, the higher the insulin resistance [25,26]. In middle-aged subjects, this fact is more pronounced in females than in males [25,26]. Insulin resistance in skeletal muscle is associated with normal aging process [24] and increased TG/HDL-C ratio [24]. Middle-aged participants had impaired insulin sensitivity [27]. The lowering of the Matsuda index, a representative of peripheral insulin sensitivity, is observed in individuals with sarcopenic obesity that is characterized by the co-occurrence of high adiposity and low muscle mass [28]. In this study, the Matsuda index was lower in middle-aged males than in young males, but there was no difference between middle-aged females and young females. The superior glucose tolerance in females than in males is primarily affected by the decline in insulin secretion with aging, rather than insulin sensitivity. In this study, the Matsuda index in middle-aged participants negatively correlated with BMI. These findings might be associated with increased impairment of peripheral insulin sensitivity and the co-occurrence of high adiposity and low muscle mass in middle-aged males.

The prevalence of high-normal FPG was higher in young males than in young females. This could be explained by HOMA-IR and HOMA-β. HOMA-IR is lower in young females than in young males, whereas HOMA-β is higher in young females than in young males.

The prevalence of IGT increased with age. This could be explained by a reduction in basal and early-phase insulin secretion, such as fasting plasma insulin and insulinogenic index. Additionally, a higher BMI and negative correlation between the BMI and Matsuda index suggest increased peripheral insulin resistance in middle-aged participants.

The prevalence of CGI, but not diabetes, was higher in middle-aged males than in middle-aged females. In this study, diabetes was strongly suspected (diabetes + CGI) in 22.8% of males and 13.5% of females. These results are comparable with the finding of the National Health and Nutrition Survey in Japan, 2019 [10]. Decreased insulin secretion increased postload glucose concentration, and elevated insulin resistance increases FPG [8]. No significant differences were found in the Matsuda index and HOMA-β, whereas the FPG, AUCg, and TG/HDL-C ratios were significantly higher in middle-aged males than in middle-aged females. These suggest that other factors besides glucose and insulin levels, such as estrogen [29,30], testosterone [31,32,33,34], lipid, and rheological parameters [13,14,35], may contribute to the difference.

The high-normal FPG (5.6–6.1 mmol/L) group was classified as the NGT group unless the 120 min postload plasma glucose increased to >7.8 mmol/L. A study of Japanese individuals showed that impaired insulin secretion and decreased insulin sensitivity already have begun in patients with FPG of 5.6–6.1 mmol/L [3]. The cut-off FPG value for the prediction of T2DM in 5 years was 5.7 mmol/L [4]. High-normal FPG levels may be associated with decreased spatial perception in young healthy adults [36], and FPG of 5.6–6.1 mmol/L was associated with endothelial dysfunction by flow-mediated vasodilation [37]. Even in young and healthy Japanese individuals who had glycemic control within the physiological range, there is a sequential decrease in insulin sensitivity and secretion [13], and cardinal rheological parameters, such as lipid parameters, reflected insulin resistance and release [14,35]. In this study, the prevalence of high-normal FPG among young males (15.2%) was significantly higher than those among young females (3.9%). Young males are thought to be more likely to develop T2DM than young females. Lifestyle habits such as emotional “comfort” eating, lack of sleep, impulsive behaviors, and selection of specific foods often lead to insulin resistance in adolescents [38]. Considering those studies, lifestyle-related obesity and lipid disorder in young males may be a good indicator of early glucose resistance.

The TG/HDL-C ratio was reported to be associated with b-cell function and insulin resistance [14,39,40,41]. In our study participants, except for young females, the Matsuda index significantly correlated with BMI and lipid parameters. Furthermore, TG was significantly higher in young males than in young females, but not in middle-aged males and middle-aged females. These findings indicated that the higher BMI in middle-aged participants attenuated females-related glucose tolerance.

Among postmenopausal females, significant differences were reported in HDL, TG, and TG/HDL ratio between healthy individuals with obesity and individuals with glucose intolerance such as low Matsuda index [42]. In this study, TG was significantly higher in males than in young females only, consistent with a previous report [43]. Among young females, the main predictors of insulin resistance were TG in normal weight, TG/HDL ratio in overweight, and TG/HDL ratio in individuals with obesity [44]. These indicated that appropriate markers vary depending on BMI.

Insulin sensitivity in premenopausal females was higher than that in young males, whereas the difference disappears in postmenopausal females partly due to 17β-estradiol [29]. In this study, we could not reveal why young Japanese females have superior glucose tolerance in terms of insulin resistance/secretion and lipid markers. To answer this question, sex difference in substrate metabolism in skeletal muscle should be considered. Although females have two-thirds the skeletal muscle mass of males and double the fat mass, lipid stores and insulin-stimulated glucose uptake in skeletal muscle is higher in females than in males. Higher levels of adiponectin and leptin are observed in females [23]. Greater body fat stores and greater lipid stores in skeletal muscle are observed in females compared with males [23]. Low-dose hormone replacement therapy with 1 mg of 17β-estradiol and 0.5 mg norethisterone to postmenopausal females with T2DM resulted in decreased FPG [30]. Consistent with previous studies, young females showed better glycemic parameters than young males and middle-aged participants. Further study is needed to elucidate 17β-estradiol-dependent carbohydrate and lipid metabolism in skeletal muscle [23].

Among males, testosterone negatively correlated with insulin secretion [31]. Androgen administration in healthy females was associated with the development of insulin resistance [32]. Higher testosterone levels in females increase the risk of metabolic syndrome and diabetes [33]. These supported our results that glucose tolerance is better in females than in males.

There was a significant difference in height and weight between young and middle-aged participants. These facts reflect the characteristics of the age-specific data for Japanese people [10]. BMI influences glucose tolerance, and it is unlikely that height or weight alone influences glucose tolerance.

This study has several limitations. First, we did not consider familial T2DM history. Second, the omission of the 90 min value might have also compromised the accuracy of our estimates of the Matsuda index, AUCg, and AUCi. Third, this study is a pure cross-sectional study and only provides indirect evidence of an increased risk for T2DM development. Fourth, participants took 75 g OGTT only once. Therefore, intraindividual variability was not assessed. Fifth, age and BMI between males and females were higher in males. These differences may affect the sex difference. Sixth, we did not measure sex hormones that may affect insulin secretion/resistance. Seventh, some middle-aged participants take drugs for treating dyslipidemia. That might weaken the relation between glucose and lipid profiles. Finally, we did not account for anthropometric measurements, food intake and exercise habits, especially the behavior of the previous day. We confirmed only 12 h of overnight fast, as it may affect insulin sensitivity or secretion.

## 5. Conclusions

This study revealed apparent sex differences in glucose tolerance, and the difference was attenuated according to age. The difference is caused by decreased β-cell function represented by fasting plasma insulin levels, HOMA-β, and increased BMI. This study showed that many middle-aged Japanese have decreased insulin secretory capacity. Many middle-aged Japanese may be at high risk of type 2 diabetes due to a decrease in basal insulin secretion.

## Figures and Tables

**Figure 1 nutrients-14-04868-f001:**
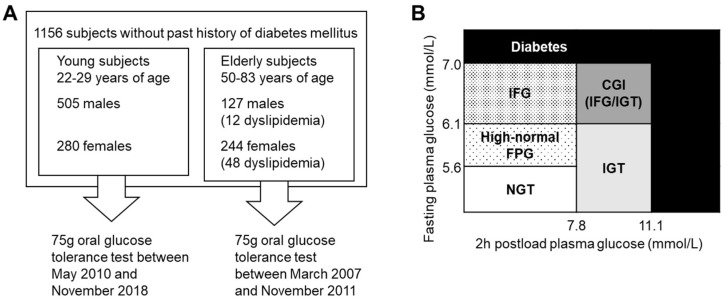
Study protocol (**A**) and classification of glucose tolerance on 75 g oral glucose tolerance test according to Japanese Clinical Practice Guideline for Diabetes 2019 (**B**). NGT, normal glucose tolerance; IFG, impaired fasting glucose; IGT, impaired glucose tolerance; CGI, combined glucose intolerance.

**Figure 2 nutrients-14-04868-f002:**
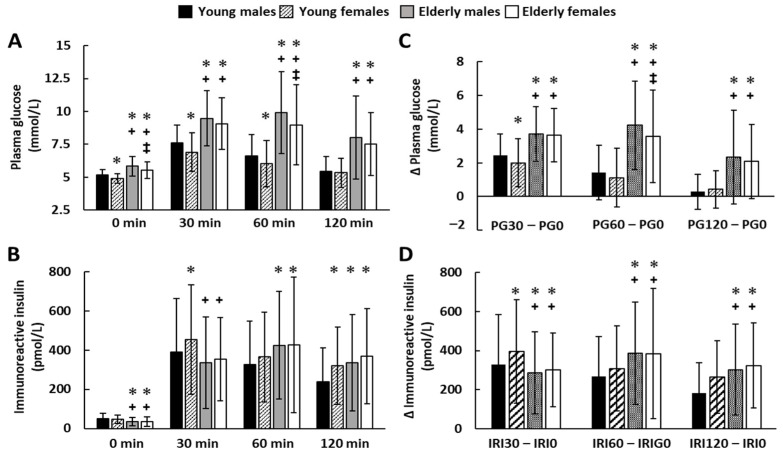
The result of 75 g oral glucose tolerance test in 505 males, 280 females, 127 middle-aged males, and 244 middle-aged females. Sequential changes in plasma glucose (**A**) and insulin (**B**) concentration in each group during 75 g oral glucose tolerance test. Differences of plasma glucose (**C**) and insulin (**D**) concentration from pre-load time in each group during 75 g oral glucose tolerance test. Data are presented as means ± SD. * *p* < 0.05 versus young males; **^+^**, *p* < 0.05 versus young females; **^‡^**, *p* < 0.05 versus elderly males, at the same time. PG0, preload plasma glucose; PG30, 30 min postload plasma glucose; PG60, 60 min postload plasma glucose; PG120, 120 min postload plasma glucose; IRI0, preload immuno-reactive insulin; IRI30, 30 min postload immunoreactive insulin; IRI60, 60 min postload immunoreactive insulin; IRI120, 120 min postload immunoreactive insulin.3.6. HOMA-β correlated with BMI, HbA1c, and TG/HDL-C.

**Table 1 nutrients-14-04868-t001:** Clinical characteristics of the study subjects.

	Young Males (n = 505)	Young Females (n = 280)	Middle-Aged Males (n = 127)	Middle-Aged Females (n = 244)
Age (year)	23.7 ± 1.7	23.3 ± 1.4	68.3 ± 7.5 *	65.1 ± 6.4 ^+,‡^
Height (cm)	172.4 ± 5.7	159.7 ± 5.7 *	163.3 ± 5.8 *	152.2 ± 5.9 ^+,‡^
Weight (kg)	65.2 ± 8.9	50.8 ± 5.8 *	62.8 ± 7.9 *	52.5 ± 7.2 ^+,‡^
BMI (kg/m^2^)	21.9 ± 2.7	19.9 ± 1.8 *	23.5 ± 2.4 *	22.6 ± 2.8 ^+,‡^
PG0 (mmol/L)	5.2 ± 0.4	4.9 ± 0.3 *	5.9 ± 0.8 *	5.6 ± 0.6 ^+,‡^
AUCg (mmol/L·min)	768 ± 130	713 ± 137 *	1078 ± 275 *	1000 ± 247 ^+,‡^
IRI0 (pmol/L)	49 ± 29	47 ± 22	35 ± 21 *	35 ± 23 ^+^
AUCi (pmol/L·min)	35,588 ± 20,485	41,019 ± 20,112 *	41,730 ± 24,395 *	42,447 ± 26,009
HbA1c (%)	5.3 ± 0.2	5.3 ± 0.2	5.5 ± 0.6 *	5.4 ± 0.4 ^+^
HOMA-IR	1.6 ± 1.1	1.4 ± 0.7 *	1.3 ± 0.8 *	1.2 ± 0.9 ^+^
HOMA-β	81.0 ± 41.5	92.7 ± 39.4 *	44.5 ± 29.9 *	48.7 ± 30.9 ^+^
Matsuda index	7.7 ± 3.9	7.2 ± 3.2	6.9 ± 3.8 *	7.1 ± 3.6
Insulinogenic index	1.4 ± 2.7	1.9 ± 6.3	0.8 ± 0.8 *	1.0 ± 2.2 ^+^
Disposition index	9.4 ± 20.4	12.0 ± 39.8	5.1 ± 6.9 *	7.8 ± 23.2 ^+^
TG (mmol/L)	2.1 ± 1.1	1.6 ± 0.7 *	3.1 ± 1.7 *	2.8 ± 1.2 ^+^
HDL-C (mmol/L)	1.5 ± 0.3	1.8 ± 0.3 *	1.5 ± 0.8	1.8 ± 0.6 ^‡^
LDL-C (mmol/L)	2.6 ± 0.7	2.5 ± 0.6	3.2 ± 0.9 *	3.2 ± 1.0 ^+^
TG/HDL-C	1.49 ± 1.02	0.95 ± 0.45 *	2.46 ± 1.88 *	1.74 ± 1.03 ^+,‡^
LDL-C/HDL-C	1.79 ± 0.7	1.45 ± 0.45 *	2.42 ± 0.98 *	1.95 ± 0.86 ^+,‡^

Data presented as means ± SD. *: *p* < 0.05 compared with young males group, ^+^: *p* < 0.05 compared with young females group, ^‡^: *p* < 0.05 compared with middle-aged males group. BMI, body mass index; PG0, preload plasma glucose; PG30, 30 min postload plasma glucose; PG60, 60 min postload plasma glucose; PG120, 120 min postload plasma glucose; IRI0, preload immunoreactive insulin; IRI30, 30 min postload immunoreactive insulin; IRI60, 60 min postload immunoreactive insulin; IRI120, 120 min postload immunoreactive insulin; TG, triglyceride; HDL-C, high density lipoprotein-cholesterol; LDL-C, low density lipoprotein-cholesterol; AUCg, area under the curve of glucose; AUCi, area under the curve of insulin; HOMA-IR, homeostasis model assessment insulin resistance; HOMA-β, homeostasis model assessment beta cell.

**Table 2 nutrients-14-04868-t002:** Classification of glucose tolerance on 75 g oral glucose tolerance test according to Japanese Clinical Practice Guideline for Diabetes 2019.

	Young Males Number (%)	Young Females Number (%)	Middle-Aged Males Number (%)	Middle-Aged Females Number (%)
NGT	414 (82.0%)	264 (94.3%) *	32 (25.2%) *	97 (39.8%) ^+,‡^
High-normal FPG	77 (15.2%)	11 (3.9) *	27 (21.3%) *	47 (19.3%) ^+^
IFG	2 (0.4%)	0 (0%)	14 (11.0%) *	16 (6.6%) ^+^
IGT	10 (2.0%)	5 (0.2%)	25 (19.7%) *	51 (20.9%) ^+^
CGI	1 (0.2%)	0 (0%)	13 (10.2%) *	8 (3.3%) ^+,‡^
Diabetes	1 (0.2%)	0 (0%)	16 (12.6%) *	25 (10.2%) ^+^
Total	505 (100%)	280 (100%)	127 (100%)	244 (100%)

Data are expressed as numbers and percent. NGT, normal glucose tolerance; FPG, fasting plasma glucose; IFG, impaired fasting glucose; IGT, impaired glucose tolerance; CGI, combined glucose intolerance. Differences between groups were compared using chi-square tests for categorical variables. *: *p* < 0.05 compared with young males group, ^+^: *p* < 0.05 compared with young females group, ^‡^: *p* < 0.05 compared with middle-aged males group.

**Table 3 nutrients-14-04868-t003:** Multiple linear regression analysis between insulin release/sensitivity indices and lipid parameters.

**Young Males**								
	**Insulinogenic Index**		**HOMA-β**		**HOMA-IR**		**Matsuda Index**	
**Parameters**	**β**	** *P* **	**β**	** *P* **	**β**	** *P* **	**β**	** *P* **
BMI			0.346	0.000	0.375	0.000	−0.099	0.035
HbA1c					0.196	0.000		
TG					−0.227	0.034	−0.144	0.003
HDL-C							0.115	0.034
TG/HDL-C					0.425	0.000		
**Young Females**								
	**Insulinogenic Index**		**HOMA-β**		**HOMA-IR**		**Matsuda Index**	
**Parameters**	**β**	** *P* **	**β**	** *P* **	**β**	** *P* **	**β**	** *P* **
BMI								
HbA1c							−0.126	0.037
TG			0.212	0.000				
HDL-C	0.130	0.032						
TG/HDL-C								
**Middle-Aged Males**								
	**Insulinogenic Index**		**HOMA-β**		**HOMA-IR**		**Matsuda Index**	
**Parameters**	**β**	** *P* **	**β**	** *P* **	**b**	** *P* **	**β**	** *P* **
BMI			0.229	0.006	0.308	0.000	−0.369	0.000
HbA1c	−0.253	0.001	−0.280	0.001				
TG								
HDL-C							0.197	0.018
TG/HDL-C	0.505	0.000	0.229	0.008	0.231	0.006		
**Middle-Aged Females**								
	**Insulinogenic Index**		**HOMA-β**		**HOMA-IR**		**Matsuda Index**	
**Parameters**	**β**	** *P* **	**β**	** *P* **	**b**	** *P* **	**β**	** *P* **
BMI			0.412	0.000	0.458	0.000	−0.408	0.000
HbA1c			−0.229	0.000			−0.171	0.002
TG								
HDL-C			0.132	0.043				
TG/HDL-C			0.291	0.000	0.180	0.001	−0.194	0.001

BMI, body mass index; TG, triglyceride; HDL-C, high density lipoprotein-cholesterol; HOMA-IR, homeostasis model assessment insulin resistance; HOMA-β, homeostasis model assessment beta cell.

## Data Availability

The datasets generated during and/or analyzed during the current study are available from the corresponding author on reasonable request. All data generated or analyzed during this study are included in this published article.

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
