# Peer review of "Age-Related Sex Differences in Glucose Tolerance by 75 g Oral Glucose Tolerance Test in Japanese"

_nutrients, 2022, doi:10.3390/nu14224868_

Round 1

Reviewer 1 Report

1. Need to consider adjusting gender comparison of metabolic biomarkers with muscle mass as confounders like muscle mass play a role in insulin sensitivity.

2. The derivation of AUC using 4 time points (0, 30, 60, 120min) may not be adequate

3. Multiple linear regression analyses table can be better presented

4. Discussion section can explain impact of gender/age-differences on prevention of Type 2 diabetes in Japanese population as currently this section is mainly descriptive. 

Author Response

We appreciate the reviewers’ comments which have been very helpful in allowing us to revise our manuscript. We attempted to address the reviewers’ questions and revised our manuscript according to the reviews’ suggestions.

Reviewer 2 Report

This study showed a series of clinical parameters, including glucose and insulin during OGTT, clinical index for glucose metabolism, and lipid profile, in 1156 participants with different sex and age. The Authors also found that the percentage in NGT/high-normal FPG/IFG/IGT/CGI/Diabetes categories is different between male/female and young/middle-aged participants. At last, the Authors did multiple regression analysis to reveal the correlation between clinical parameters and beta cell function index. It was concluded that the superior glucose tolerance in young women may be due to higher insulin secretion potential.

While the clinical data is solid, some improvement could be made in terms of the way to analyze and interpret the data. Also, some deeper discussion is encouraged.

1. The HOMA-IR and HOMA-b models are relatively old with some limitations. There is an updated HOMA2 model, which could be considered. Details can be found in these articles (https://doi.org/10.2337/diacare.27.6.1487, https://doi.org/10.2337/diacare.21.12.2191)

2. In Table 1, there was a significant difference in height between young and middle-aged participants. What is the reason? Does it have any indirect effects to the glucose metabolism?

3. The basal insulin secretion (fasting level) is lower in middle-aged individuals than young individuals. This is opposite to the previous publication (e.g. reference 5). Is this a common phenomenon? This should be discussed.

4. In Table 2, the significant difference was shown by the p<0.05. However, unlike other data which has a mean±SD in each group, the percentage data does not. How is the p value calculated?

5. The structure of the Result and Discussion could be improved. Currently, there are too many parameters/index but without a clear focus to bring them together. 

6. Discussion could go deeper to provide more useful information. E.g. in Line 210-212, "The indices of insulin resistance/secretion and lipid profiles well correlated among individuals, but not among young women.", what is the possible reason for it? Please go through the whole Discussion part and try to discuss the possible mechanisms behind the difference in the parameters.

7. It would be good to emphasize the novelty of this study in the Abstract, Introduction, and Conclusion parts. 

8. Some indexes used in the manuscript were not solid, leading to strange conclusions. E.g. in Line 233-235, the Authors stated that HOMA-IR may be misleading. If this is the case, the Authors may choose not to use it in the manuscript, or choose a better index with less misunderstanding.

Author Response

(The authors gave the same response as above.)

Round 2

Reviewer 1 Report

In the current form, study is difficult to understand with many acronyms and multiple comparisons across age group and gender that is not clearly articulated. Discussion section definitely needs more work in relating the results to novelty of findings. The style of writing needs improvement. I would suggest choosing the most significant findings and elaborate instead of attempting to discuss all the findings.

In section 3.5 and Figure 2, absolute glucose and insulin values were used to compare between gender and age. Did the authors consider using the change in baseline values instead? That would help normalize the differences in fasting values and allow comparison of the change in values above fasting. 

In table 3, why are there missing values for each of the parameters? The readers would assume that all indices/metabolic parameters would be collected? Please consider including and analzying the complete data where all parameters were collected. 

Author Response

Answer to Reviewer1, round2

We appreciate the reviewers’ comments which have been very helpful in allowing us to revise our manuscript. We attempted to address the reviewers’ questions and revised our manuscript according to the reviews’ suggestions.

  1. In the current form, study is difficult to understand with many acronyms and multiple comparisons across age group and gender that is not clearly articulated. Discussion section definitely needs more work in relating the results to novelty of findings. The style of writing needs improvement. I would suggest choosing the most significant findings and elaborate instead of attempting to discuss all the findings.

Answer: We are very grateful for your very kind comments. In this study, lipid metabolism markers were also analyzed in addition to 75 g OGTT. We provided a number of indicators for this purpose. As a result, although lipid markers also reflect glucose tolerance, BMI, fasting insulin, and HOMA-b, which can be calculated from blood glucose levels, are good indicators. Therefore, lipid metabolism markers cannot be removed to demonstrate this fact. Please understand. We described the most significant findings of this study in the conclusion. We added the following sentences. Please confirm.

Line 256-258.

Although lipid markers also reflect glucose tolerance, BMI, fasting insulin, and HOMA-b, which can be calculated from blood glucose levels, are good indicators.

Line 400-405.

This study revealed apparent sex differences in glucose tolerance, and the difference was attenuated according to age. The difference is caused by decreased b-cell function represented by fasting plasma insulin levels, HOMA-b and increased BMI. This study showed that many middle-aged Japanese have decreased insulin secretory capacity. Many middle-aged Japanese may be at high risk of type 2 diabetes due to a decrease in basal insulin secretion.

  1. In section 3.5 and Figure 2, absolute glucose and insulin values were used to compare between gender and age. Did the authors consider using the change in baseline values instead? That would help normalize the differences in fasting values and allow comparison of the change in values above fasting.

Answer: Thank you for your very valuable comment. Through additional studies, we were able to reconfirm the high insulin secretion ability of young women. Following your instructions, we analyzed change from baseline values of glucose and insulin to 30, 60, 120 minutes. Interestingly, sex differences were apparent in young subjects, however, it was attenuated with aging. Sex differences in glucose tolerance among young people may be caused by basal and early-phase insulin secretion. We added Figure 2C and D. We added the following sentences. Please confirm.

Line 201-213.

Figure 2C shows the difference between the plasma glucose level before and after the glucose load. In young participants, sex differences were observed only at 30 minutes after glucose load and before load. In middle-aged subjects, sex differences were observed only at 60 minutes after glucose load and before load. The difference between the plasma glucose level before and after the glucose load was greater in middle-aged subjects than in young subjects. Figure 2D shows the difference between the plasma insulin level before and after the glucose load. In young participants, sex differences were observed only at 30 minutes after glucose load and be-fore load. In middle-aged subjects, there was no difference between men and women in post- and pre-dose insulin concentrations. The difference between the plasma insulin level 30 min after and before the glucose load was smaller in middle-aged subjects than in young subjects. The difference in insulin concentration between 60 min and preload, and between 120 min and preload, was greater in middle-aged subjects than in young subjects.

Line 269-273.

This study supports this theory. The difference in insulin concentrations 30 min after and before the glucose load were higher in young women than in men, and were otherwise at similar levels. The difference in plasma glucose levels 30 minutes after glucose load and before load was lower in young women than in men and was otherwise at the same level. This sex difference disappears in middle-aged men and women.

  1. In table 3, why are there missing values for each of the parameters? The readers would assume that all indices/metabolic parameters would be collected? Please consider including and analzying the complete data where all parameters were collected.

Answer: We conducted multiple linear regression analysis by stepwise method by using IBM SPSS Statistics version 25 (IBM Corp., Armonk, NY, USA). In the analysis procedure, excluded variables were not described in the results. We could not find the statistical value of the excluded variables. Therefore, the statistical value of excluded variables was blank. Please understand our situation.

Reviewer 2 Report

The Authors answered most of my questions. However, there are still some unsolved issues.

In the response to my previous question 3 (and in Line 273-275), the Authors stated that "These facts could explain the lower fasting plasma insulin levels in middle-aged participants in this study". However, I still do not understand why middle-aged participants had lower fasting insulin levels in this study. If it is a characteristic of Japanese or Asian people, the Authors may need to include other publications describing this characteristic to support it. 

The other issue is, the Authors stated that "The lower fasting plasma insulin levels caused lower HOMA-IR in middle-aged participants, as misleading as good insulin sensitivity" (in Line 276-277) without explaining why they believe it is "misleading". Although it is common knowledge that insulin sensitivity gradually decreases along aging, whether it is the case in this study (or in Japanese people) needs more support by data or reference. Also, the statement "Therefore, HOMA-IR were not suitable for middle-aged participants" (Line 277-278) seems very strong, which requires reference to support.

Regarding Table 3, all of the correlation seems not strong, although significant. According to a guideline of the BMJ (https://www.bmj.com/about-bmj/resources-readers/publications/statistics-square-one/11-correlation-and-regression), 0-0.19 is regarded as very weak, 0.2-0.39 as weak, 0.40-0.59 as moderate, 0.6-0.79 as strong and 0.8-1 as very strong correlation. Most of the values in Table 3 indicate that the correlation is very weak or weak. The weak correlation may need to be stated in the whole paper, including the abstract, to avoid any misleading.

In Line 288-293, the Authors stated that "Middle-aged participants had impaired insulin sensitivity. The lowering of the Matsuda index......These facts support our results that the superior glucose tolerance in women was apparent in young, but attenuated in middle-aged women". However, in Table 1, there was no difference between young and middle-aged women regarding the Matsuda index. The Authors need to be careful about such discussion.

Author Response

Answer to Reviewer2 round2

We appreciate the reviewers’ comments which have been very helpful in allowing us to revise our manuscript. We attempted to address the reviewers’ questions and revised our manuscript according to the reviews’ suggestions.

Comment 1. In the response to my previous question 3 (and in Line 273-275), the Authors stated that "These facts could explain the lower fasting plasma insulin levels in middle-aged participants in this study". However, I still do not understand why middle-aged participants had lower fasting insulin levels in this study. If it is a characteristic of Japanese or Asian people, the Authors may need to include other publications describing this characteristic to support it.

Answer: Fasting plasma insulin levels in our study participants were the same as other studies which investigated middle-aged Japanese. These reports are already listed in the references, as [3] and [7]. Therefore, the lower fasting plasma insulin levels in middle-aged participants in this study was a characteristic of Japanese or Asian people.

Therefore, we discarded the following sentences, (Line 268-270) “Although no participants were diagnosed with T2DM in the past or received antidiabetic medication, 25.2% of middle-aged men and 39.8% of middle-aged women were classified as NGT by 75 g OGTT.”

We added following sentences, please confirm.

Line 270-277. Fasting plasma insulin levels were 35±21 pmol/L and the same in 32 middle-aged men classified as NGT and in all middle-aged men. Similarly, fasting plasma insulin levels were 35±23 pmol/L in 97 middle-aged women classified as NGT and 35±21 pmol/L in all middle-aged women. Fasting plasma insulin levels in this study participants were the same as other studies which investigated middle-aged Japanese [3, 7]. Therefore, the lower fasting plasma insulin levels in middle-aged participants in this study was a characteristic of Japanese or Asian.

Comment 2. The other issue is, the Authors stated that "The lower fasting plasma insulin levels caused lower HOMA-IR in middle-aged participants, as misleading as good insulin sensitivity" (in Line 276-277) without explaining why they believe it is "misleading". Although it is common knowledge that insulin sensitivity gradually decreases along aging, whether it is the case in this study (or in Japanese people) needs more support by data or reference. Also, the statement "Therefore, HOMA-IR were not suitable for middle-aged participants" (Line 277-278) seems very strong, which requires reference to support.

Answer: We appreciate the comment. We agree with this opinion. “Therefore, HOMA-IR were not suitable for middle-aged participants” is an overestimation.

Therefore, we discarded these descriptions. We also revised Discussion based on this point.

We discarded the following sentences.

Line 140-143. “which was misinterpreted as good insulin sensitivity. Therefore, HOMA-IR were not suitable for middle-aged men. Therefore, we performed multiple regression analysis without HOMA-IR in middle-aged men (Table 3).”

Line 166-168. “which was misinterpreted as good insulin sensitivity. Therefore, HOMA-IR were not suitable for middle-aged women. Therefore, we performed multiple regression analysis without HOMA-IR in middle-aged women (Table 3.)”

Line 280-281. “as misleading as good insulin sensitivity. Therefore, HOMA-IR were not suitable for middle-aged participants.”

Line 298-299. “These facts support our results that the superior glucose tolerance in women was apparent in young, but attenuated in middle-aged women.”

We added the following sentences and HOMA-IR of middle-aged men and women in Table 3, please confirm.

Line 137-138. Compared to young men, in middle-aged men, BMI increased and the Matsuda index and HOMA-b decreased, while unexpectedly HOMA-IR decreased.

Line 162-164. Compared to young women, in middle-aged women, BMI increased and HOMA-b decreased, while unexpectedly HOMA-IR decreased.

Line 294-298. In this study, the Matsuda index was lower in middle-aged men than in young men, but there was no difference between middle-aged women and young women. The superior glucose tolerance in women than in men is primarily affected by the decline in insulin secretion with aging, rather than insulin sensitivity.

Comment 3. Regarding Table 3, all of the correlation seems not strong, although significant. According to a guideline of the BMJ (https://www.bmj.com/about-bmj/resources-readers/publications/statistics-square-one/11-correlation-and-regression), 0-0.19 is regarded as very weak, 0.2-0.39 as weak, 0.40-0.59 as moderate, 0.6-0.79 as strong and 0.8-1 as very strong correlation. Most of the values in Table 3 indicate that the correlation is very weak or weak. The weak correlation may need to be stated in the whole paper, including the abstract, to avoid any misleading.

Answer: We appreciate the comment. We agree with this opinion. We revised “significantly” to very weakly, weakly or moderately according to a guideline of the BMJ. Please confirm.

We revised Abstract, Line 28; Results, Line 208 and 210-222; and Discussion, Line 244

Comment 4. In Line 288-293, the Authors stated that "Middle-aged participants had impaired insulin sensitivity. The lowering of the Matsuda index......These facts support our results that the superior glucose tolerance in women was apparent in young, but attenuated in middle-aged women". However, in Table 1, there was no difference between young and middle-aged women regarding the Matsuda index. The Authors need to be careful about such discussion.

Answer: We appreciate the comment. We agree with this opinion. We revised Discussion according to this comment.

We discarded the following sentences.

Line 298-299. “These facts support our results that the superior glucose tolerance in women was apparent in young, but attenuated in middle-aged women.”

We added the following sentences, please confirm.

Line 294-298. In this study, the Matsuda index was lower in middle-aged men than in young men, but there was no difference between middle-aged women and young women. The superior glucose tolerance in women than in men is primarily affected by the decline in insulin secretion with aging, rather than insulin sensitivity.
